# Protocol for a Randomized Controlled Trial to Enhance Executive Function via Brief Mindfulness Training in Individuals with Internet Gaming Disorder

**Zhilin Chen**[1,2‡], **Jie Ge**[2,3‡], **Quan Gan**[1,2,4], **Yu Fu**[1,2], **Zhuangfei Chen**[1,2*]

**1** Medical School, Kunming University of Science and Technology, Kunming, China, **2** Brain Science and Visual Cognition Research Center, Medical School of Kunming University of Science and Technology, Kunming, China, **3** Students Counseling and Mental Health Center, Kunming University of Science and Technology, Kunming, China, **4** Faculté de médecine, Université Paris-Saclay, Le Kremlin-Bicêtre, France

‡ These authors contributed equally to this work and share first authorship on this work.
* chen.zhf@outlook.com

## Abstract

### Background

Internet Gaming Disorder (IGD) is characterized by uncontrolled gaming behavior, leading to emotional distress, neglect of academic or life responsibilities, and damage to interpersonal relationships, all of which have serious negative impacts on individuals and society. IGD has been associated with neuropsychological impairments, especially executive function deficits, and emotional difficulties. Mindfulness interventions have been indicated to improve executive functions to varying degrees in individuals with IGD. The purpose of this study is to investigate whether the three subcomponents of executive function (i.e., inhibition, updating, and shifting) are impaired in individuals with internet gaming disorder, as well as to examine the effectiveness of a brief mindfulness intervention on executive function.

### Methods

A total of 82 individuals diagnosed with IGD and 40 non-addicted gamers will be recruited for this study. These participants will be evenly divided into an intervention group and a control group at a ratio of 1:1. The intervention group will undergo a 7-day mindfulness training program focusing on breathing meditation, while the control group will receive progressive muscle relaxation training. Both groups' outcomes will be assessed at seven different time points. Primary outcome measures will include electroencephalography (EEG) data (band power, functional connectivity, source localization, and N2/P3 amplitudes), behavioral metrics (response times and accuracy from psychological experimental paradigms), physiological indices (specifically heart rate variability), self-reported measures (game craving and mindfulness levels, executive function performance, and impulsivity emphasizing inhibitory control). Secondary outcome measures will encompass

**Data availability statement:** No datasets were generated or analysed during the current study. All relevant data from this study will be made available upon study completion.

**Funding:** This research was supported by the National Natural Science Foundation of China (NSFC)(Nos. 32060196, 32460211, 82360271 and 82201597), Yunnan Fundamental Research Projects (202401AT070332), Joint Funds for Medical Specialization of KUST (KUST-KH2023002Y), Yunnan Ten Thousand Talents Plan Young and Elite Talents Project (YNWR-QNBJ-2018-027, YNWR-QNBJ-2018-056), and The innovation team of stress and disorder in nervous system in Yunnan (202305AS350011). The funders had no involvement in the study design, data collection and analysis, decision to publish, or manuscript preparation. Financial support was provided in the form of research materials.

**Competing interests:** This study was supported by the National Natural Science Foundation of China (NSFC), Yunnan Fundamental Research Projects, Joint Funds for Medical Specialization of KUST, Yunnan Ten Thousand Talents Plan Young and Elite Talents Project and Innovation team of Stress and disorder in nervous system in Yunnan. The funders had no role in study design, data collection and analysis, decision to publish, or preparation of the manuscript. The financial support was provided in forms of research materials. The authors have declared that no competing interests exist.

anxiety, stress, positive and negative affect, sleep quality, and other indicators such as demographic information, physical and mental health status, and the Big Five personality traits.

## Discussion

This study aims to explore the efficacy of a brief mindfulness intervention on executive function impairments in IGD individuals and to elucidate its underlying neural mechanisms. It is anticipated that the findings will contribute to more targeted intervention strategies for executive function research, offering novel insights into the treatment of IGD and related cognitive dysfunctions. This study is expected to explore the effectiveness of brief mindfulness intervention on IGD and its underlying brain functional mechanisms, particularly providing more targeted intervention strategies for improving executive functions in this population.

## Trial registration

Trial registration number: ChiCTR2400081509, registered on March 4th 2024. Protocol Version1.0.

## 1. Introduction

### 1.1.  IGD and cognitive function

In 2013, Internet Gaming Disorder (IGD) was included in the Diagnostic and Statistical Manual of Mental Disorders, Fifth Edition (DSM-5) as a condition "warranting further study" [1]. In 2019, when the World Health Assembly adopted the 11th revision of the International Classification of Diseases (ICD-11) [2], gaming disorder was categorized under disorders due to addictive behaviors. Despite variations in nomenclature and criteria, the core issue remains excessive gaming. The estimated global prevalence of IGD is 8.8% among adolescents and 10.4% among young adults [3]. Studies have demonstrated that IGD can lead to adverse consequences in adolescents, including academic decline, sleep deprivation [4], relationship problems, loneliness, suicidal ideation [5], aggression, depression [6], and social withdrawal.

Research indicates that IGD is associated with neuropsychological impairments, especially executive function deficits, and emotional difficulties [3,7]. Based on dual-process models within addiction frameworks [8,9], cognitive-behavioral models [10], and the I-PACE model [11,12], it is evident that both affective and cognitive mechanisms play crucial roles in the development and maintenance of internet gaming disorder. Reduced executive control over gaming-related cravings is considered a central feature of IGD [10], and cue-induced cravings also require regulation by executive functions, making this area a critical focus in IGD research. Miyake et al. [13] proposed one of the most influential models of executive functions, identifying three subprocesses through latent variable analysis: inhibition, updating, and shifting. Inhibition refers to the ability to consciously suppress automatic or dominant responses; updating involves continuously refreshing the contents of working memory, including imagery and rule information relevant to the task; shifting pertains to the flexible transition between different tasks, operations, or mental sets. The primary behavioral characteristic of IGD is excessive gaming behavior despite awareness of negative consequences, which is underpinned by functional impairments. Functional magnetic resonance imaging (fMRI) studies on individuals with IGD have revealed abnormalities in brain structures such

as the dorsolateral prefrontal cortex, anterior cingulate cortex, and insula [14–19], regions closely linked to executive function. However, research on executive function in IGD has primarily focused on the inhibition component, with less attention given to shifting and updating abilities, particularly regarding cognitive flexibility, which remains controversial in the context of excessive gaming [20–22]. Prolonged video game playing appears to enhance abilities related to sensorimotor coordination, which can contribute to improved shifting functions [21,22]; however, the observed weakening of inhibition abilities raises concerns about a corresponding decline in shifting functions within the unified executive function system. A preliminary investigation into the shifting functions of individuals addicted to online games [20] found that neurophysiological changes indicated enhanced shifting functions at the neural level, whereas behavioral results did not show such improvements. A comprehensive investigation of all three subcomponents of executive function is essential for a deeper understanding of the cognitive mechanisms related to IGD.

## 1.2. Mindfulness and behavioral addiction

Mindfulness involves deliberately focusing one's attention on experiencing thoughts, emotions, and bodily sensations, and cultivating an attitude of awareness, acceptance, and non-judgment toward present-moment experiences [23]. The benefits of mindfulness can largely be summarized as improvements in attention, emotion, and cognition, with attention and executive function considered core mechanisms [24]. Lutz et al. [25] provided a theoretical framework for focused meditation, describing mindfulness practices that include: (1) maintaining attention on a momentary or specific object; (2) detecting mind wandering or attentional dispersion (i.e., attention monitoring); (3) disengaging attention from distractors and re-focusing on the moment or object; (4) nonjudgmental evaluation of distractions. Gallant [26] posits that diverting attentional resources away from distractions and subsequently re-focusing them on the current task involves and trains inhibitory control and shifting abilities but does not directly engage or enhance updating capacity. However, mindfulness may indirectly influence updating through stress reduction or enhanced relaxation, thereby enhancing attention to the current task [27].

The core clinical symptoms of addiction include heightened motivation (craving), impaired self-control (impulsivity and compulsion), emotional dysregulation (negative emotions), and increased stress responses [28]. Mechanisms involved in mindfulness meditation practices, such as mindful reappraisal, attentional and emotional regulation, and metacognitive awareness, may have direct therapeutic effects in reducing maladaptive cognitions and increasing positive reappraisals, thus decreasing negative emotions, stress, impulsivity, and gaming craving [29].

## 1.3. Mindfulness in the intervention of IGD

### 1.3.1. Mindfulness, emotional regulation, and impulsivity in IGD. Stress serves as a potent predictor of addiction, while negative emotions are a prevalent factor triggering craving and relapse. When these states overwhelm an individual who feels powerless to escape them, the motivation to use addictive substances becomes more pronounced [30]. Following mindfulness interventions, there is an observed increase in activation of the anterior cingulate cortex (ACC) and medial prefrontal cortex (mPFC), regions intimately involved in emotional regulation [28]. Numerous studies have highlighted the effectiveness of mindfulness-based interventions in alleviating stress and enhancing resilience in substance addiction [31,32].

Increased impulsivity is a hallmark of individuals with substance use disorders, correlating with poorer treatment outcomes and higher rates of relapse. Research indicates that

individuals with IGD exhibit greater decision-making impulsivity, favoring smaller immediate rewards over larger delayed ones [33]. After undergoing mindfulness-based therapy, IGD individuals demonstrate reduced levels of decision-making impulsivity, with decreases in delay discounting significantly correlated with lower severity of IGD [33].

Studies on addiction underscore the pivotal role of craving [34,35]. Craving facilitates the development and sustenance of addictive behaviors [36]. Despite awareness of adverse consequences, impaired control over craving is linked to addictive behaviors. Mindfulness meditation enhances attention and self-regulation, potentially diminishing craving [29]. Prior research has documented reductions in craving and substance use among drug users following mindfulness meditation [37], with lasting impacts on craving reduction [38].

**1.3.2.  The therapeutic potential of brief mindfulness interventions.**  Mindfulness, as an efficacious therapeutic approach, has been extensively applied in addiction and relapse prevention, showing promise in mitigating craving and maladaptive cognitions in individuals with internet gaming disorder [39]. Research suggests that mindfulness may improve prefrontal cortex function [40–42], a brain region critical for various aspects of executive function, including inhibitory control, mental set shifting and working memory [43,44]. Studies indicate that mindfulness may improve specific aspects of executive function in IGD patients, as evidenced by reduced addiction severity and enhanced brain connectivity, which correlates with certain biochemical changes. However, further research is needed to establish clear causal relationships between mindfulness practice, changes in executive function, and biochemical alterations [45].

Research indicates that the prevalence of IGD among young adults is significantly higher than that of adolescents (10.4% vs. 8.8%) [3]. College students are more likely to engage in problematic gaming behaviors compared to other young people, as they may have more free time and minimal supervision from schools and parents, allowing them to play video games intensely [46]. An American report suggests that 25% of people suffer from smartphone addiction [47]. Given China's large population base, it possesses the world's largest number of smartphone users, particularly among college students [48]. Earlier studies suggested that the severity of problematic smartphone use (PSU) exhibits similar traits to other forms of internet overuse, such as IGD. Additionally, the high accessibility and diverse range of smartphone applications may contribute to this overuse [49]. The escalating prevalence of IGD and its detrimental effects underscore the urgent need for interventions targeting young people, especially college students, to avert more severe socio-psychological repercussions.

It is well-known that long-term mindfulness training can lead to participant dropout [50], constrained by conditions such as prolonged duration and face-to-face facilitation. As a result, demand for brief self-help practices is growing. Existing research confirms that a single 30-minute mindfulness intervention can ameliorate smartphone addiction [51]. Moreover, studies indicate that short-term mindfulness can variably improve the three subcomponents of executive function [52]. Follow-up studies reveal a relapse rate of 70.6% in IGD individuals one year post-discharge [53], highlighting the challenge in sustaining treatment effects for relapse-prone conditions like IGD and underscoring the need for innovative therapeutic approaches. There is an urgent need to develop new treatment methods. Existing literature systematically describes the treatment and advantages of cognitive behavioral therapy, family therapy, pharmacotherapy, and physical therapy for individuals with gaming addiction [54,55]. Mindfulness, as an emerging approach in addiction treatment, is still in the exploratory stage of application in gaming addiction interventions and is receiving increasing attention from researchers.

Studies on substance addiction have shown that mindfulness is a long-term intervention strategy [56], suggesting that if individuals with IGD persist in mindfulness meditation training over the long term, it may help reduce the likelihood of relapse.

Currently, there is a lack of sufficiently in-depth research on whether the three subcomponents of executive function are impaired in individuals with internet gaming disorder, and whether mindfulness interventions can improve executive function and elucidate related neural mechanisms in both IGD populations and healthy controls.

Therefore, we hypothesize that brief mindfulness interventions reduce impulsivity and gaming craving in individuals with IGD by improving executive functions (including inhibitory control, cognitive flexibility, and working memory), thereby alleviating their addiction symptoms (Fig 3).

This study aims to conduct a randomized controlled trial to examine the effects of a one-week focused-breathing mindfulness meditation on executive function. By addressing the time-consuming nature of mindfulness practices, this approach seeks to contribute to more targeted intervention strategies in executive function research. While conceptual articles have outlined potential mechanisms of mindfulness treatment for IGD, empirical research on the therapeutic mechanisms linked to brief mindfulness practices remains scarce.

## 2.  Materials and methods

### 2.1.  Study design and participants

This trial adopts a parallel randomized controlled trial design. Given the mixed but promising literature, some studies support a moderate effect of brief mindfulness interventions on certain aspects of executive function, while others report inconsistent findings [52]; we aim to explore whether the improvement in executive function under the brief mindfulness intervention condition is superior to that of the control group for individuals with IGD.

Participants will be recruited through a variety of methods to ensure a sufficient and representative sample size, including distributing posters, conducting on-site presentations, as well as advertisements during university class breaks, all while adhering to specified inclusion and exclusion criteria (Table 1). Individuals exhibiting IGD will be

Table 1.  Inclusion and exclusion criteria.

| Inclusion Criteria | Exclusion Criteria |
|---|---|
| (1) Right-handedness;<br>(2) Normal vision or corrected-to-normal vision;<br>(3) Age between 18 and 40 years;<br>(4) Good mental and physical health (PHQ-9 total score < 20, GAD-7 total score < 11);<br>(5) For individuals with internet gaming disorder (IGD): Internet Addiction Test (IAT) score ≥ 50, DSM-5 score ≥ 5, gaming history exceeding 2 years, and gaming time of more than 14 hours per week;<br>(6) For recreational gamers: IAT score < 50, DSM-5 score < 5. | (1) Individuals with asthma, epilepsy, or a history of mental illness;<br>(2) Those who have taken steroid medications within the past three months;<br>(3) Individuals who have practiced meditation, yoga, Tai Chi, or Qi Gong for more than 20 hours in the past year or throughout their lifetime, attended meditation or yoga retreats, or participated in any meditation courses;<br>(4) Individuals unsuitable for electroencephalogram (EEG) testing due to metal implants, severe head injuries, contact dermatitis, or silicone allergies;<br>(5) Those with a PHQ-9 total score ≥ 20 or a GAD-7 total score ≥ 11;<br>(6) Individuals with other addictions (such as smoking, alcohol, or gambling);<br>(7) Individuals with strong adherence to a particular religious belief that would interfere with their ability to participate in the required meditation exercises;<br>(8) Individuals currently participating in similar trials or other neurophysiological studies. |

Note: PHQ (Patient Health Questionnaire); GAD (Generalized Anxiety Disorder scale); DSM-5 (Diagnostic and Statistical Manual of Mental Disorders, Fifth Edition).

identified using a two-step process. First, participants scoring above 50 on a modified Internet Addiction Test (IAT) [57,58] will be selected. Second, these individuals will undergo psychiatric interviews based on DSM-5-TR (Text Revision) criteria for IGD diagnosis (requiring at least six of nine criteria for significant severity) [59]. Co-occurring psychiatric conditions will be assessed with the Chinese Version of the Mini-International Neuropsychiatric Interview (M.I.N.I.) [60,61].

A total of 122 participants are planned to be enrolled, comprising 82 individuals with internet gaming disorder and 40 recreational gamers. The SPIRIT guidelines have been followed (see S2 File for details), which provide a detailed timeline for each study component. The study was registered with the Clinical Trials Registry (registration number: ChiCTR2400081509, S3 File). Moreover, we recruit recreational gamers to ensure the integrity of our experimental design. The control group should account for additional variables such as familiarity with the game and gender. Among healthy control groups that do not meet the addiction criteria, the presence or absence of gaming experience can lead to different experimental results, which is particularly important when the experiment involves game-related stimuli. Recent studies have begun to include recreational users of online games in the control groups. These recreational players constitute a significant proportion of gamers. By comparing game addicts with recreational users, we can obtain more reliable addiction-related markers by balancing game familiarity [62,63]. Informed consent will be obtained from all participants before the study commences. Participants will have the right to withdraw from the study without any obligation to provide a reason to the researchers. During the follow-up period, participants will receive monthly intensive training sessions and will track their self-guided practices online every two weeks. This schedule is based on previous research showing that monthly follow-ups effectively monitor sustained benefits while balancing participant engagement and feasibility [64,65]. The study will take place in the participants' current living environment at the university.

## 2.2. Ethics and dissemination

The study has been approved by the medical ethics committee of Kunming University of Science and Technology on October 30th, 2023 (approval number: KMUST-MEC-211). Written informed consent will be obtained from each participant prior to participation in accordance with the Declaration of Helsinki and its later amendments or comparable ethical standards. Any modifications to the protocol will be reported to the ethics committee and the trial registration site. Changes to methods and outcomes after trial commencement must be transparently reported, justified, and documented in the trial registration.

## 2.3. The status and timeline

Due to the multi-stage nature of the data collection process in this study, we expect to complete participant recruitment by March 2025. As of submission, recruitment is ongoing and conducted in batches. We anticipate that preliminary data analysis will conclude by the end of May 2025, with longitudinal results expected by the end of June 2026. The publication plan will include sections on research design, statistical analysis of panel data, psychological data analysis, and conclusions.

## 2.4. Sample size calculation

Considering that our primary outcomes include multiple indicators, we referenced previous studies involving scale scores [66], behavioral measures [67–69], electroencephalography [70], and heart rate variability [71]. The sample size calculation will primarily target the effect

size of executive function, especially in studies related to the Stroop task, given its pivotal role in our research. By focusing on this most challenging primary outcome and considering both the effect size and variance associated with the Stroop task as key factors, we will ensure a sufficiently robust sample size. This approach will not only guarantee adequate statistical power for analyzing all primary outcomes but also enhance the reliability and validity of our findings.

We will estimate the sample size using the pwr package in R software. Here, we set $\delta(\Delta)$tot = 125, corresponding to an expected reduction of 125 milliseconds in the reaction time of the experimental group following the intervention. We will set the significance level ($\alpha$) at 0.05 and the power ($\beta$) at 0.80. To estimate Cohen's d, the formula is cohen'd = ($\delta(\Delta)$/SDestimate), where the SDestimate = 250 based on studies with similar populations [68]. Therefore, the relevant command will be pwr.t.test(d = cohen_d, sig.level = 0.05, power = 0.80, type = "paired", alternative = "two.sided"), yielding an estimated sample size of 34. Hence, for the intervention group of internet gaming disorder participants, a sample size estimate of no fewer than 34 people will be required; considering a natural dropout rate of 20%, the sample of internet gaming disorder participants in the intervention group should not be less than 41 individuals. A ratio of 2:1 will be used to include 20 recreational gamers in the intervention group. A total of 61 participants will be required for the intervention group. In order to maintain appropriate comparability, we will recruit an additional 61 participants with matching demographic profiles and equivalent levels of IGD severity for the control group. Thus, the study demands a cumulative participation of 122 individuals in total.

## 2.5. Group allocation

This study is a parallel randomized controlled trial (RCT) designed following the recommendations of the Consolidate Standards of Reporting Trials (CONSORT) and will be reported accordingly (S4 File). After obtaining informed consent, taking gender and IGD severity (i.e., IAT and DSM-5 score) into account, random numbers generated by a random sequence generator website (https://www.random.org/) will be used to allocate participants to the intervention or control group at a ratio of 1:1. Allocations will be obtained by email from staff who were not involved in other parts of the trial. Participants in the intervention group will undergo seven consecutive days of mindful breathing meditation training, lasting 1.5 hours per day. Simultaneously, participants in the control group will receive progressive muscle relaxation (PMR) training, which will be conducted over a period of 7 days, consistent with the intervention group. Data will be collected at seven time points: baseline (-T2, -T1; averages will be taken to reduce random errors), post-intervention (T0), and follow-up (T1: one week after data collection ends; T2: one month later; T3: four months later; T4: six months later; and T5: one year later). The study flow is illustrated in Figs 1–3, Fig S1 and S1 File. Given the substantial body of literatures supporting the moderate effect of brief mindfulness interventions on executive function, particularly in the inhibition and updating subfunctions [72], this trial aims to demonstrate the superiority of such interventions for individuals with IGD compared to a control group.

## 2.6. Intervention program and control design

**2.6.1. The intervention group.** The intervention group will be trained by a certified psychotherapist (JG) with extensive experience in mindfulness practices from the University's Students Counseling and Mental Health Center. Data collection for intervention group will be accomplished by XP and QC. During the 7-day training period (intervention, INT), participants will practice designated content through check-ins on the Ruixin Meditation App

(https://sj.qq.com/appdetail/com.carson.mindfulnessapp) after each session. The after-class exercises do not specify a time or location; they only need to be completed on the app on the same day. Considering that brief mindfulness training might not produce observable long-term effects due to limited exposure time, following settings from existing research, after the initial post-intervention data collection (T0), there will be monthly online group sessions (2 hours) organized. The content of these sessions will mirror the 7-day training, followed by another 7 consecutive days of post-session check-ins via the app, during which participants are encouraged to practice independently and complete at least one check-in per day. The app will monitor and enable users to record their self-practice time for each phase.

As depicted in the specific intervention scheme, the intervention group will undergo Brief Mindfulness Training (BMT) involving mindful breathing meditation (Table 2).

**Mindful breathing meditation training** (Mindful Breathing or MB): The training content is based on established concepts and practices recognized in mindfulness literature [73]. The program centers around mindfulness meditation, which includes focusing on selected objects (such as the body or breath), monitoring mental activities, noticing distractions, and cultivating a non-judgmental attitude towards one's own experiences (i.e., calmness). The emphasis is placed on body scanning, mindful breathing, breathing space, seated meditation, mindful walking, and mindful stretching. This structure aims to cultivate a deep understanding and practice of mindfulness, enabling participants to integrate this practice into their daily lives and respond to stress more mindfully.

| | STUDY PERIOD | | | | | | | |
|---|---|---|---|---|---|---|---|---|
| | Enrolment | Allocation | Post-allocation | | | | | Close-out |
| TIMEPOINT* | -T₂ | -T₁ | T₀ | T₁ | T₂ | T₃ | T₄ | T₅ |
| **ENROLMENT:** | | | | | | | | |
| **Eligibility screen** | X | | | | | | | |
| **Informed consent** | X | | | | | | | |
| **Baseline assessment** | X | X | | | | | | |
| **Allocation** | | X | | | | | | |
| **INTERVENTIONS:** | | | | | | | | |
| *[Brief mindfulness training]* | | | ●——● | | | | | |
| *[Relaxation training]* | | | ●——● | | | | | |
| **ASSESSMENTS:** | | | | | | | | |
| *[Demographic data]* | X | | | | | | | |
| *[Primary outcome variables]* | X | X | X | X | X | X | X | X |
| *[Secondary outcome variables]* | X | X | X | X | X | X | X | X |
| *[Daily records during intervention]* | | | X | X | X | X | X | X |

**Fig 1. SPIRIT schedule of enrolment, interventions and assessment.** Note: T2, T1, baseline (complete two baseline measurements to get average); T0, post-intervention; T1- T5, 1-week, 1-month, 4-month, 6-month,1-year follow-up.

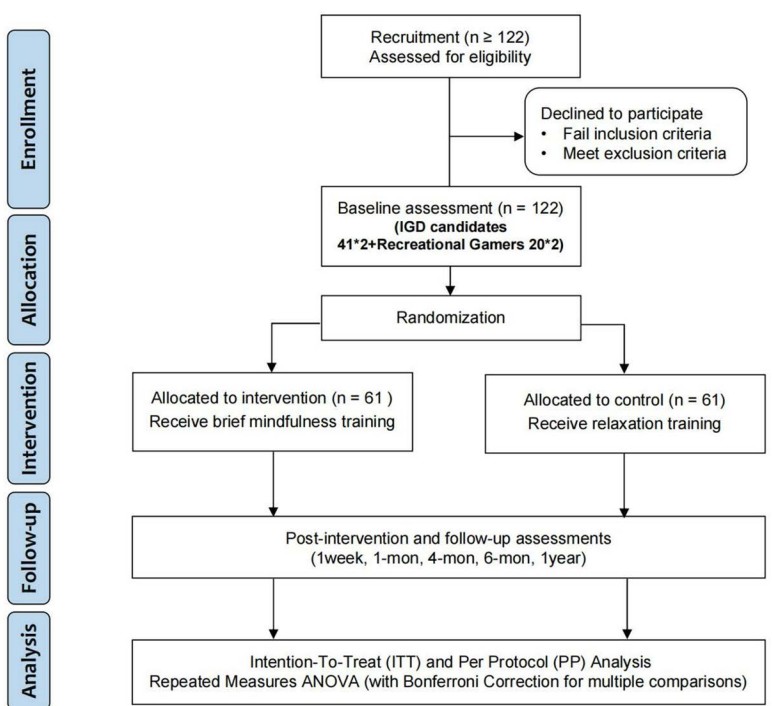

**Fig 2. SPIRIT diagram.** Note: Completed through the Chinese version of the questionnaire, scales, and additional self-rated questionnaires, and self-designed questionnaires. IGD: Internet Gaming Disorder.

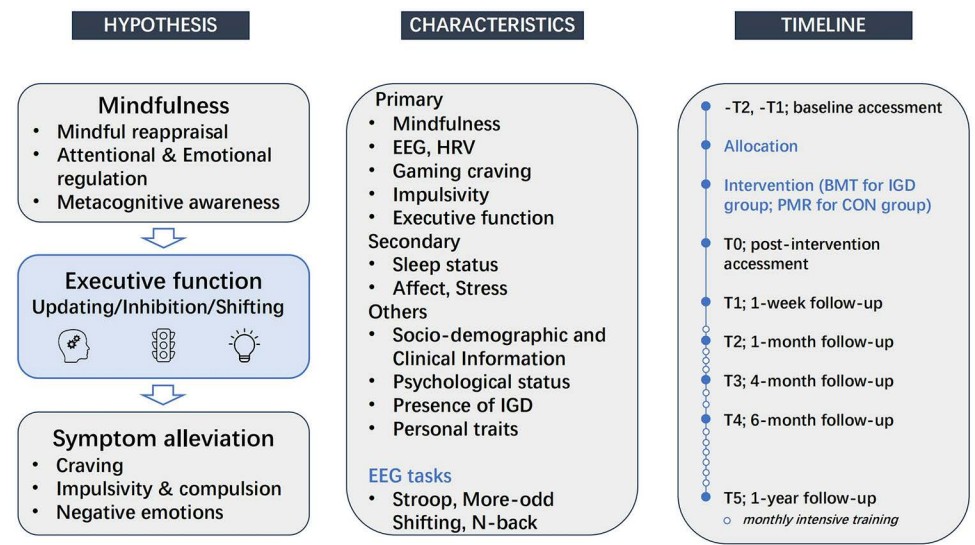

**Fig 3. Diagram of research hypotheses, assessment indicators, and measurement timeline.** Note: HRV refers to heart rate variability.

**Table 2.  Overview of the intervention strategies for both groups.**

|  | **Brief Mindfulness Training (BMT)** | **Progressive Muscle Relaxation (PMR)** |
|---|---|---|
| **Day1** | **Introduction to Mindfulness:**<br>- Overview of mindfulness and the attitudes associated.<br>- Establishing connection within the group.<br>- Brief introduction to mindfulness practice.<br>- Mini body scan.<br>- Story of the Old Well and Pebbles (Explores course intentions through imagery: "Why are we here? Why practice mindfulness?").<br>- Raisin exercise (Cultivates awareness by observing a raisin, enhancing sensory focus). | Active Music-Assisted Progressive Relaxation (Kangning music treatment, KN-MT) combined with the PMR technique from Zhihua's "Journey to Rediscover Inner True Love" workshop, along with interactive sharing. |
| **Day2** | **Perception and Creative Response:**<br>- Check-in sharing about mindfulness experiences.<br>- Introduction to stress response patterns.<br>- Observing changes in reaction patterns.<br>- Opening meditation.<br>- Body scan.<br>- Simple yoga poses | Passive Music-Assisted Progressive Muscle Relaxation provided by the KN-MT, also incorporating Zhihua's PMR technique from the same workshop, accompanied by interactive sharing. |
| **Day3** | **Finding an Anchor in the Present Moment:**<br>- Sitting meditation for 15 minutes.<br>- Body scan for 30 minutes.<br>- Mindful yoga for 10 minutes.<br>- Mindfulness sharing.<br>- Introduction to the practice of recording pleasant life events (Bean Counting Exercise ("Spot a joyful moment? Move a bean from your left to your right pocket to keep track and enjoy the sensation").<br>- Mindful interpersonal communication (silent listening and full sharing). | Repeated sessions of the Active Music-Assisted Progressive Relaxation Method (KN-MT) with PMR from Zhihua's workshop, including interactive sharing. |
| **Day4** | **Mindfulness in Interpersonal Relationships:**<br>- Sitting meditation for 15 minutes.<br>- Mindful yoga for 30 minutes.<br>- Mindful interpersonal communication (six people silently share an orange for 10 minutes).<br>- Reading "Life Is Now" together, focusing on the orange meditation.<br>- Introduction to loving-kindness practice.<br>- Mindfulness sharing and Q&A. | Multiple instances of Passive Music-Assisted Progressive Muscle Relaxation facilitated by the KN-MT, using the PMR technique taught by Zhihua Zhang, with interactive sharing incorporated. |
| **Day5** | **Mindful Response to Life Stress:**<br>- Group reading of the first two sections of "Life Is Now." (Thich Nhat Hanh's work emphasizes mindfulness for emotional understanding and deeper connections).<br>- Awareness of emotions, expression and listening.<br>- Mindful walking.<br>- Sharing and listening in small groups about pleasant and unpleasant life events.<br>- Practice of the breathing space.<br>-Mountain Meditation Practice (Cultivates stability and strength by envisioning oneself as a resilient mountain). | Additional rounds of the Active Music-Assisted Progressive Relaxation Method (KN-MT) paired with Zhihua's PMR method, followed by interactive sharing. |
| **Day6** | **Outdoor Walking Meditation, Deeply Observing Self and Nature:**<br>- Outdoor walking meditation starting at 6 AM, including a simple explanation, silent meditation, and one-hour outdoor walk meditation, followed by a return to the classroom for a summary. | Further sessions of Passive Music-Assisted Progressive Relaxation conducted by the KN-MT, guided by Zhihua's PMR approach, with interactive sharing included. |

*(Continued)*

**Table 2.** (Continued)

| | Brief Mindfulness Training (BMT) | Progressive Muscle Relaxation (PMR) |
|---|---|---|
| Day7 | **Sustaining Mindfulness in Daily Life:**<br>- Mindful sitting meditation.<br>- Mindful body scan.<br>- Simple yoga.<br>- Group closure, video recording, Two-person sharing, reflections and decisions on practicing mindfulness, playing back the walking meditation video, group loving-kindness and blessings, integrating mindfulness into the present. | Consecutive repetitions of the Active Music-Assisted Progressive Relaxation Method (KN-MT) alongside PMR instruction from Zhihua Zhang, complemented by interactive sharing. |
| Post-class exercises (Ruixin app - tracks practice time) | Mindful Breathing + 5-Day Mindfulness Introduction:<br>- A five-day intensive introduction to mindfulness meditation focusing on breathing. | Relaxation music to facilitate the overall experience, organized online by dedicated staff. |

**2.6.2. The control group.** The control group will be led by another psychology instructor (ND) at the same time, who will provide course management and audio practice, while the intervention group receives mindfulness training. While data collection for control group will be conducted by XA and ZL. Participants in control group will engage in Progressive Muscle Relaxation (PMR) exercises (Table 2). As a traditional design, PMR has been used as the control condition to study the efficacy and neural mechanisms of mindfulness meditation in IGD [66]. Previous research observed slight gaming craving reduction with PMR, potentially due to a placebo effect. However, PMR group did not show significant changes in executive control network connectivity, and the difference from the mindfulness intervention group remained substantial [45]. This highlights mindfulness' unique efficacy, as PMR uses a distinct relaxation mechanism.

**Progressive Muscle Relaxation Training** (PMR): PMR was originally defined by Jacobson and involves voluntary, continuous, and systematic stretching and relaxation of various muscle groups [74]. This exercise draws an individual's attention to their skeletal muscles, promoting whole-body relaxation. The audio-guided practice begins with instructions for slow, rhythmic breathing. It then guides participants to notice tension in their muscle groups (face and neck; chest, shoulders, upper back, and abdomen; as well as the right and left thighs, calves, and feet), allowing these groups to fully relax. Subsequently, participants are guided through active tensing and relaxing of muscles. Finally, a countdown is used to guide participants into complete relaxation. The PMR training sessions feature content from the Himalayas audio platform (https://www.ximalaya.com), designed to promote deep relaxation and stress relief through a combination of active and passive muscle relaxation techniques, guided by soothing music and expert instruction. Interactive sharing allows participants to connect and learn from each other's experiences, enhancing the therapeutic benefits of the training.

## 2.7. Participant withdrawal and management

Besides the previously mentioned withdrawal mechanism, participants attending fewer than 5 intervention sessions will be excluded from the study. Any adverse events, such as experiencing significant emotional discomfort (e.g., anxiety, sadness), psychological challenges due to triggered memories (especially for those with a history of trauma), physical discomfort from

maintaining still postures, or social isolation from an excessive focus on inner experiences, may lead to early termination of their participation. Participants can request to suspend their involvement, and if no adverse events occur after evaluation, they will be transferred to the future intervention group, with their data excluded from the final analysis.

## 2.8. Strategies for consistent batch interventions

To ensure consistency in batch recruitment and intervention, we will implement standardized procedures (e.g., structured interview guides), initial and ongoing personnel training, strict timetables, and monitoring mechanisms (periodic reviews, real-time tracking of key indicators, participant and staff feedback, quality control checks, detailed documentation, manual reviews at critical points, ethical oversight, and peer review) to enhance data reliability and research quality.

## 2.9. Data collection

Participants are scheduled to come to the laboratory at their convenience for data collection sessions. Each session involves filling out questionnaires and undergoing electroencephalogram (EEG) recordings. Recognizing the impact of neural excitability on cognitive functions, participants will be instructed to avoid gaming, caffeine consumption, and smoking three hours prior to EEG data collection.

The study's data collection timeline is summarized, detailing the exact timing of measures taken (Table 3; Fig 3, Fig S1). Primary outcomes will encompass a comprehensive set of indicators. EEG data will be analyzed to examine band power, functional connectivity, and source localization, as well as the average amplitudes of N2 and P3 event-related potential (ERP) components. Behavioral measures will focus on response times and accuracy from psychological experimental paradigms. Additionally, heart rate variability (HRV) will be included as a physiological index. Self-reported measures will include assessments of game craving and mindfulness levels, as well as evaluations of executive function and impulsivity, particularly focusing on inhibitory control. Secondary outcomes include anxiety, stress, positive and negative affect, and sleep quality, all of which may be influenced by mindfulness levels. Additional personal data such as demographics, physical and mental health status, and Big Five personality traits will also be collected. To ensure questionnaire reliability, all questionnaires will be filled out on paper initially. These responses will then be transcribed into electronic formats for data analysis. All inventories/questionnaires are in Chinese version and has been validated (Table 3).

For the collection of EEG and behavioral data, the study utilizes E-prime 3.0 for paradigm programming and execution. Simultaneously, EEG data will be recorded using the ActiveTwo BioSemi system, and real-time heart rate will be tracked by a Huawei Honor Band. The 64-channel electrode cap adheres to the international 10-20 system, operates at a sampling frequency of 512 Hz, and maintains electrode impedance below 30 KΩ. This comprehensive setup ensures high-quality data acquisition for detailed analysis. To participants who failed to finish data collection mentioned above, we will send reminder messages through WhatsApp in a timely manner, especially for those that need to be completed in the laboratory.

## 2.10. Executive functional tasks

The executive function tasks will be conducted to assess different components of executive function, including inhibition, shifting, and updating. Here's a breakdown of the tasks. A single EEG recording session will last approximately 40 minutes and will include the following components: resting state (5 minutes), executive function task 1 (practice + formal, 15

minutes), mindfulness (5 minutes), executive function task 2 (formal, 10 minutes), and resting state (5 minutes) (Fig S1). The selection of executive function tasks will be based on similar studies [96,97] but the parameter settings will be modified according to the pilot study of the current project. Specifically, the duration of stimulus presentation and the inter-stimulus interval will be determined based on the maximum response time of participants in the pilot

**Table 3. Evaluated variables.**

| Variables | | Measures | Timepoints |
|---|---|---|---|
| **Primary outcome** | **Trait-, State-mindfulness** | Five Facet Mindfulness Questionnaire (FFMQ) [74,75] | -T2, -T1; T0, T1, T2, T3, T4, T5 |
| | | State Mindfulness Scale [76] | |
| | **Electroencephalographic (EEG) and behavioral data** | Behavioral and Electroencephalogram (EEG) data | |
| | | | Heart Rate Variability |
| | **Gaming craving-related metrics** | Average daily gaming time (self-recorded by participant) | |
| | | | Internet Addiction Test (IAT) [57,58] |
| | | | Gaming Craving Visual Analogue Scale [77] |
| | | | Gaming Craving Short Questionnaire (adapted from QSU-Brief) [78,79] |
| | **Impulsivity** | Barratt Impulsiveness Scale [80,81] | |
| | **Executive function** | Adolescent Executive Functioning Scale (EFS-A) [82] [a] | |
| **Secondary outcome** | **Sleep status** | Self-Rating Scale of Sleep (SRSS) [83] | -T2, -T1; T0, T1, T2, T3, T4, T5 |
| | **Affect, Stress** | Depression Anxiety Stress Scales (DASS-21) [84] | |
| | | | Positive and Negative Affect Schedule (PANAS) [85,86] |
| **Others** | **Socio-demographic and clinical information** | Demographics including name, gender, handedness [87,88], gaming history duration, average weekly gaming time [b], psychiatric diagnosis or history of mental health problems | -T1 |
| | **Psychological status** | Patient Health Questionnaire-9 (PHQ-9) [89,90] | |
| | | | Generalized Anxiety Disorder-7 (GAD-7) [91, 92] |
| | **Presence of IGD** | Diagnostic and Statistical Manual of Mental Disorders, Fifth Edition (DSM-5) nine diagnostic criteria [59] | |
| | | | Gaming history, average weekly gaming time [c] |
| | **Personal traits** | Chinese Big Five Personality Questionnaire [93], attachment style [94], self-compassion [95] | |

Note: a. Participants included undergraduates; scale showed good reliability and validity; b. Average weekly gaming time serves as an auxiliary criterion for determining addiction (as seen in the inclusion criteria); c. Average daily gaming time is used to explore the changes in gaming duration among gamers before and after the intervention (effect of mindfulness intervention).

study. The test sequence will include the Stroop task, the More-odd Shifting task, and the N-back task. Executive function task 2 will be conducted without practice and directly will enter the formal experiment to reduce practice effects and prevent participant fatigue from prolonged task duration.

**Stroop Task:** This task will evaluate the inhibition component of executive function. After instructions, the black cross will appear at the center of the screen for 500 ms as the fixation point, followed by a colored Chinese character stimulus for up to 2000 ms. The character will disappear upon key press, followed by a blank white screen for 500 ms, then the cycle will repeat. The characters will be the words for "red," "green," "blue," and "yellow," written in those colors. There will be 16 possible combinations. Congruent conditions will have matching color and meaning, while incongruent conditions will not. The ratio of congruent to incongruent trials will be 1:3, with 12 congruent and 36 incongruent trials, totaling 48 trials. Before the formal experiment, there will be 16 practice trials. Greater differences in reaction times between incongruent and congruent conditions will indicate poorer inhibitory control.

**More-odd Shifting Task:** The task will assess the shifting component of executive function. Similar to the Stroop Task setup, participants will perform sub-tasks involving size judgment, odd/even judgment, and mixed size/odd-even judgment. Red numbers will be for size judgment (less than 5 press 'f', greater than 5 press 'j'), green numbers will be for odd/even judgment (odd press 'f', even press 'j'). The mixed judgment task will combine both rules, with red numbers for size and green numbers for odd/even. Practice and formal trial counts will vary, but smaller differences in average reaction times between switching and non-switching tasks will suggest better shifting abilities.

**N-back Task:** This task will evaluate the updating component of executive function. Following instructions, participants will see a black cross, then single letters will appear for up to 2000 ms. After a key press, a 500 ms blank screen will appear before the next letter. Two sub-tasks will be included: 1-back and 2-back. For 1-back, participants will judge if the current letter matches the previous one; for 2-back, participants will judge whether the current letter matches the letter two steps back. Each task will include $45 + n$ judgments, with 15 practice trials before the formal experiment. Matched and unmatched stimuli ratios will be 1:2. The N-back task will challenge working memory and updating capabilities.

## 2.11. Data management

Prior to statistical analysis, all participant data will be uniformly anonymized. Additionally, secure measures such as data encryption and other robust storage methods will be employed to protect the confidentiality and integrity of the data.

We will adhere rigorously to standardized protocols for each type of data acquisition method, ensuring consistency and accuracy in our data collection process. All source data will be securely stored on designated platforms, with redundant backups maintained on specialized, offline hard drives or external drives dedicated to data analysis. For data housed on third-party online platforms, we will promptly download and securely store these files to safeguard against cyber threats, enhance device protection, prevent data breaches, maintain robust backup practices, and conduct regular vulnerability assessments to prevent unauthorized access. These tasks will be executed by personnel who have undergone specialized training to ensure high-quality data entry and management. Furthermore, to enhance data integrity, we will implement a dual-entry verification system and conduct range checks on data values to ensure accuracy and consistency throughout the datasets.

For missing data, we will adopt the assumption that they are missing at least at random (MAR), a hypothesis that will be validated through appropriate analyses. To account for instances of unanticipated missing data, we will employ multiple imputation (MI) techniques to improve the precision of our findings. Furthermore, we will incorporate secondary variables to assist in refining the estimation of any missing values. Sensitivity analyses will be conducted to assess the effects of different MI models, including both missing at random and missing not at random scenarios, to investigate how missing data might influence the overall results of the study.

Data analysis will be conducted after all stages of data collection, without interim analysis. Study data will be available to qualified researchers post-study, via the corresponding author. All information and data collected from participants during the study will be handled confidentially. We will not share any identifiable information with parties external to the research team without explicit participant consent. In public reports of the study findings, participant identities will remain anonymous.

Considering the limited timeframe and the low known risk to participants in the current study, forming a Data Monitoring Committee is deemed unnecessary. However, the research team will remain vigilant in monitoring the situation and will reassess the need for establishing such a committee should circumstances change.

## 2.12. Safety considerations

Dr. Farias outlines adverse events related to meditation, grouped into mental, physical, and neurological/cognitive categories [98]. Mental issues include anxiety, depression, and psychosis. Physical problems consist mainly of tension, pain, and gastrointestinal issues. Neurological/cognitive effects involve disorganized thinking and sensory changes.

Our study includes weekly questionnaires to monitor adverse events, with immediate support provided when needed. For EEG data collection, we halt if participants experience discomfort, prioritizing their health and consent. This protocol ensures a safe and responsive approach to managing adverse events throughout the study. Upon completing the trial, if a participant's stress level reaches a clinical threshold, they will be offered active intervention through the on-campus psychological services. If necessary, referrals to specialized healthcare facilities will be provided for additional support.

## 2.13. Data analysis

In our study, data analysts will be blinded to the intervention allocation to minimize bias during data analysis.

**Scales**: Between-group comparisons will be conducted using independent samples t-tests, while within-group comparisons will employ paired t-tests. Repeated measures ANOVA will be applied to analyze data collected at different time points across multiple sessions. SPSS 20.0 (IBM SPSS Statistics, IBM Corp., Armonk, NY, USA) and Rversion 4.4.1 (R Foundation for Statistical Computing, Vienna, Austria) will be utilized for these analyses. Effect size will be calculated using Cohen's d or partial eta-squared values for primary and secondary outcomes to assess treatment impact. Data yielding a $p < 0.05$ will be considered statistically significant. The effect size, along with its 95% confidence interval (CI), will be reported.

**EEG Data:** EEG data will be processed using Matlab R2013b (MathWorks in Natick, Massachusetts, USA), EEGLab 13.0.0b (https://sccn.ucsd.edu/eeglab) software, and customized

scripts. Power Spectrum and Functional Connectivity Analysis: EEG data will be used to calculate relative power in six EEG frequency bands (δ: 1-4Hz, θ: 4-8Hz, α: 8-13Hz, low-β: 13-20Hz, high-β: 20-30Hz, low-γ: 30-48Hz) at both single electrode and ROI levels. Phase Locking Value (PLV) and Phase Lag Index (PLI) will be calculated using the FieldTrip tool- box. Source localization analysis will be employed to explore functional changes in subcortical regions.

**ERP Data**: N2 and P3 components will be selected based on previous studies. Epochs will have a duration of 1000 ms, with a baseline of 200ms pre-stimulus onset. N2, a negative component primarily elicited in the frontocentral region, will be analyzed at frontal mid- line electrodes. The time window for N2 is 250-350ms post-stimulus onset. P3, a positive component appearing in the parietal-central area, will be analyzed at fronto-parietal electrodes, with a time window of 300-400ms post-stimulus onset. To minimize Type I error rates, we will not analyze individual electrodes but rather the average amplitude values within selected time windows. All ERP component analyses will be completed using ERPLAB.

In summary, following the examination of band power, functional connectivity, and source localization in EEG data, ERP components will be extracted. HRV will serve as an index for evaluating physiological responses.

Finally, we will conduct both an Intention-To-Treat (ITT) analysis and a Per Protocol (PP) analysis for this trial. The ITT analysis helps preserve the random assignment of par- ticipants and reduce bias, while the PP analysis provides insights based on participants who completed the intervention as intended. We considerattrition rates as a critical indicator of the study's integrity and will carefully record and analyze them. This approach will allow us to evaluate the effects of mindful breathing meditation on executive function from both perspectives.

As mentioned in the data management section, to address missing data, we will employ multiple imputation techniques to enhance our findings' precision. Differences in sample size at various time points due to attrition rates may complicate our analysis. Therefore, we will implement repeated measures ANOVA cautiously, accounting for sample size changes to maintain statistical power. To mitigate the Type I error rate, we will apply Bonferroni correc- tion during post hoc testing to adjust the significance level for each comparison.

## 3. Discussion

The research on the impact of mindfulness interventions on executive functions primarily targets populations such as those with depression, ADHD, and elderly individuals with degenerating brain functions, with less focus on individuals with IGD. In this study, we will control for baseline differences in the sample by excluding other addictions (smok- ing, alcohol) and selecting two groups of game users (addicted and recreational) to determine the true impact of gaming addiction on executive functions. Both groups will undergo mindfulness interventions and muscle relaxation training, allowing us to inves- tigate the influence of mindfulness on planning, decision-making, attention shifting, and the underlying mechanisms related to negative emotions, stress, impulsivity, and cravings for games.

For brain function evaluation, EEG tasks will measure participants before and after the inter- vention, while negative emotions will be assessed through scales. Additionally, our study will conduct heart rate variability measurements during resting and brief mindfulness periods to investigate both acute [99,100] and chronic [101,102] effects of mindfulness based on the State Mindfulness Scale.

## Strengths and future implications

This study is expected to enrich research on mindfulness and executive functions in individuals with IGD. The beneficial outcomes may be as follows:

Firstly, this research aims at enhancing our understanding of executive functions by providing insights into how mindfulness interventions improve various components, such as inhibitory control, working memory, and flexibility. Additionally, it offers a comprehensive evaluation of long-term effects through multiple assessments, which help uncover the relationship between mindfulness practices and executive function components. Furthermore, this study seeks to provide a scientific basis for developing new treatment strategies for IGD, while expanding the knowledge base for future mindfulness studies. Moreover, by incorporating a range of outcome measures—such as EEG data, behavioral metrics, and mindfulness levels—along with secondary outcomes like anxiety and sleep quality, the research aims for a holistic assessment.

Finally, it sheds light on individual differences, such as age and gender, that influence the effectiveness of mindfulness interventions, while also assessing potential psychological and social health benefits. Ultimately, this research aims to deepen our understanding of the effects of mindfulness on executive functions in individuals with IGD, thereby providing practical guidance for improving their quality of life.

The potential for brief mindfulness interventions could be integrated into various clinical settings, such as primary care clinics, community health centers, and mental health practices. For example, healthcare providers can incorporate brief mindfulness exercises into regular appointments, and community organizations can offer group sessions. Additionally, training programs for clinicians and educators can be developed to ensure they are equipped to deliver these interventions effectively. This approach would make the benefits of mindfulness more accessible to a broader population, including those with IGD, and support their executive function improvements in real-world settings.

Furthermore, future studies could incorporate well-designed web-based treatments with effective strategies to reduce attrition rates [103]. Although we implemented appropriate measures in our current design to prevent participant dropout, such as sending reminder notifications to enhance engagement and ensuring the anonymity of data management and analysis to improve compliance, there are still various strategies from previous online studies that can be referenced.

For instance, it has been shown that providing a coach for advice and support helped one pre-post study achieve a 70% retention rate [104], and ensuring privacy and convenience in accessing the intervention are critical factors for participant engagement [104–106]. Future research should explore increasing engagement through mechanisms like incentives and interactive methods, such as motivational interviewing via videoconferencing [107,108], to enhance personal interaction and improve retention rates for online addictions [109].

## Limitations

As to limitations, in the first place, there is a lack of consensus on theoretical models to differentiate between the two populations, which will result in the absence of validated diagnostic criteria and cut-off points. Many studies rely on self-reported measures of IGD for inclusion criteria; however, self-reporting will not be adequate for clinical diagnosis and may yield a high number of false positives [110]. Therefore, identifying potential biomarkers is essential to enhance detection and intervention methods. Subsequently, a fully double-blind randomized controlled trial will not be feasible due to the likelihood of participants recognizing and

communicating with each other, making it difficult for trainers to mask the interventions provided. Additionally, the extensive time required to carry out experiments will likely lead to an excessively large sample size, making it impossible to complete the study; thus, a placebo control will not be established in this research.

This study will employ a diverse range of paradigms for cognitive function measurement, which will complicate cross-study comparisons. However, multi-level cognitive function assessments will enhance the interpretative depth of the findings. Specifically, while standard executive function tasks can assess executive dysfunction related to addiction, they cannot diagnose it. Consequently, the current lack of specific biomarkers for IGD limits our understanding and treatment options. In addition, daily gaming time will be self-reported by participants, and the lengthy duration of the experiment may introduce bias and data loss. Lastly, as the mindfulness intervention model will require personalized settings, achieving uniformity in intervention duration and content is likely to be challenging, potentially leading to issues with systematic fidelity assessment and replication during implementation.

## Supporting information

**S1 File. Copy of the study protocol.**
(DOCX)

**S2 File. SPIRIT checklist.**
(PDF)

**S3 File. All items from Registration Data Set.**
(DOCX)

**S4 File. CONSORT checklist.**
(PDF)

**Fig S1. Study design and procedure.** Note: The upper portion indicates the overall study design. The middle portion indicates the timeline for assessment of scales and EEG at baseline and post-intervention, also follow-up time-points. Brief mindfulness and relaxation training was administered to intervention group and control group after recruitment.Baseline assessment (-T2, -T1) was finished before intervention. Audio guidance was delivered for everyday's practice for BMT group. There is no other practice and intervention for control group after training. The lower portion indicates modules of psychological and behavioral assessments during the baseline and 5 follow-up sessions. Executive function measurement was implemented via EEG tasks including Stroop, More-odd Shifting, N-back. INT: intervention, INST:monthly intensive training (red triangle); rs-EEG: resting-state EEG; BMT: brief mindfulness training; VAS: Visual analogue scale; SMT: state-mindfulness.
(TIF)

## Author contributions

**Conceptualization:** Jie Ge, Zhuangfei Chen.

**Data curation:** Zhilin Chen, Jie Ge.

**Funding acquisition:** Yu Fu, Zhuangfei Chen.

**Investigation:** Zhilin Chen, Zhuangfei Chen.

**Methodology:** Jie Ge, Quan Gan, Yu Fu, Zhuangfei Chen.

**Project administration:** Quan Gan, Yu Fu, Zhuangfei Chen.

**Resources:** Yu Fu, Zhuangfei Chen.

**Supervision:** Jie Ge, Quan Gan, Yu Fu, Zhuangfei Chen.

**Writing – original draft:** Zhilin Chen, Jie Ge.

**Writing – review & editing:** Jie Ge, Quan Gan, Yu Fu, Zhuangfei Chen.

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
