## [Decision Letter · Decision Letter 0]

7 Nov 2024

PONE-D-24-33010Protocol for a Randomized Controlled Trial to Enhance Executive Function via Brief Mindfulness Training in Individuals with Internet Gaming DisorderPLOS ONE

Dear Dr. Chen,

Thank you for submitting your manuscript to PLOS ONE. After careful consideration, we feel that it has merit but does not fully meet PLOS ONE’s publication criteria as it currently stands. Therefore, we invite you to submit a revised version of the manuscript that addresses the points raised during the review process.

Please see the reviewers' comments.

We look forward to receiving your revised manuscript.

Kind regards,

Marc N. Potenza

Academic Editor

PLOS ONE

2. Thank you for stating the following financial disclosure: [This study was supported by the National Natural Science Foundation of China (NSFC) (Nos. 32060196, 82360271 and 82201597), Yunnan Fundamental Research Projects (202401AT070332), Joint Funds for Medical Specialization of KUST (KUST-KH2023002Y), Yunnan Ten Thousand Talents Plan Young and Elite Talents Project (YNWR-QNBJ-2018-027, YNWR-QNBJ-2018-056) and The innovation team of stress and disorder in nervous system in Yunnan Province (202305AS350011).]. Please state what role the funders took in the study. If the funders had no role, please state: "The funders had no role in study design, data collection and analysis, decision to publish, or preparation of the manuscript." If this statement is not correct you must amend it as needed. Please include this amended Role 

Additional Editor Comments (if provided):

Reviewers' comments:

Reviewer's Responses to Questions

**Comments to the Author**

1. Does the manuscript provide a valid rationale for the proposed study, with clearly identified and justified research questions?

Reviewer #1: Yes

Reviewer #2: Yes

Reviewer #3: Yes

2. Is the protocol technically sound and planned in a manner that will lead to a meaningful outcome and allow testing the stated hypotheses?

Reviewer #1: Partly

Reviewer #2: Yes

Reviewer #3: Yes

3. Is the methodology feasible and described in sufficient detail to allow the work to be replicable?

Reviewer #1: Yes

Reviewer #2: Yes

Reviewer #3: No

4. Have the authors described where all data underlying the findings will be made available when the study is complete?

Reviewer #1: Yes

Reviewer #2: Yes

Reviewer #3: No

5. Is the manuscript presented in an intelligible fashion and written in standard English?

Reviewer #1: Yes

Reviewer #2: Yes

Reviewer #3: Yes

6. Review Comments to the Author

You may also provide optional suggestions and comments to authors that they might find helpful in planning their study.

Reviewer #1: Line 226: the randomization is unclear. State clearly how these 82 individuals with internet gaming addiction and 40 recreational gamers are allocated into the intervention and control group.

Line 281: Several primary outcomes will be studied. Is the sample size calculated based on executive function sufficient to adequately support the analysis of the other primary outcomes?

Line 292 for the progressive muscle relaxation training: To mention it will be held 7 days similarly with the intervention group.

Line 294: The sentence ‘following the intervention’ is to be omitted.

Line 302: A separate subtitle to indicate intervention and control group respectively.

Information on the person who provide the training and do the data collection is to be provided.

Page 22 Table 3: Information on whether all these inventories/questionnaires are in Chinese version and has been validated is to be provided.

Data management on missing data if any is to be mentioned.

Line 476: Usually repeated measures ANOVA post hoc tests able to handle both within-group and between-group comparisons. Any reason why independent t test and paired t test are used?

Line 478: The name of the effect size is to be provided.

Line 478: Complete citation for SPSS 20.0 and R software including publisher name and version is to be provided.

Line 478-479: Effect size with 95%CI to be stated.

The accepted statistically significant level is to be provided.

Line 496-499: To state clearly whether both intent-to-treat analysis and per protocol analysis will be conducted. If there are differs ‘n’ at various time points (due to attrition rates or missing data), using repeated measure ANOVA becomes problematic.

When performing repeated measures ANOVA with multiple time points, there is a risk of inflating the type I error rate (false positives) due to multiple comparisons. Thus, it is important to impose Bonferroni correction during post hoc testing to adjust the significance level for each comparison.

Figure 2: the analysis section requires revision.

References did not conform to the journal format.

Reviewer #2: The manuscript presents a well-conceived study protocol that is likely to contribute valuable insights into the impact of mindfulness on executive function in individuals with IGD. By exploring the effects of mindfulness training on executive function in individuals with IGD, the authors aim to bridge a gap in current intervention methods and provide new therapeutic insights. The manuscript clearly defines both primary (e.g., EEG data, executive function) and secondary (e.g., anxiety, stress) outcomes, ensuring that both cognitive and psychological dimensions are covered. This comprehensive evaluation can provide a holistic understanding of the intervention's impact on IGD.

I only have some minor concerns for this well-documented manuscript.

1. Longitudinal Follow-up Justification: While the study’s longitudinal design is commendable, the rationale for the choice of follow-up intervals (e.g., one week, one month, etc.) could be strengthened. Are these intervals based on previous findings regarding the duration of mindfulness effects? Providing a clearer justification for these time points could clarify the expected temporal effects of mindfulness training.

2. Control Condition (Progressive Muscle Relaxation): The choice of progressive muscle relaxation (PMR) as a control condition is appropriate for differentiating between general relaxation and mindfulness-specific effects. However, the manuscript could benefit from a brief discussion on whether PMR is expected to influence executive function or gaming cravings to any degree, as this could affect the interpretation of results.

3. Terminology and Consistency: Ensure consistency in the terminology, such as using “Internet Gaming Disorder” (IGD) consistently instead of alternating with "gaming addiction" to prevent ambiguity.

4. Future Implications: The manuscript could be further strengthened by a more explicit discussion of how the findings may be translated into clinical practice, particularly for interventions beyond university-based or app-based contexts.

Reviewer #3: This manuscript presented an interesting RCT protocol that will examine the effectiveness of a brief mindfulness intervention on executive function in individuals that game (i.e., individuals with IGD and individuals without). I have provided comments on specific areas for improvement below.

Major issues:

1. Page 4, Rows 69-70: In the abstract, it may be beneficial to specify what you will be looking at in the EEG and what the behavioral metrics will specifically be measuring. Please also consider applying these changes to the main text of the manuscript (e.g., Page 21, Rows 370-371).

2. Page 6, Row 113: I would recommend being slightly cautious of the phrasing here, as there is no consensus in the literature on the primary diagnostic characteristic of IGD. However, I agree that “excessive gaming behavior despite awareness of negative consequences” is one of the key characteristics of IGD.

3. Page 7, Row 119: Given the focus on shifting and updating abilities throughout the manuscript, a brief description of each might be helpful to readers.

4. Page 7, Rows 123: Before introducing the mindfulness sections, please provide more context on current IGD treatment by briefly discussing the most common intervention types for IGD, those that have shown promise, and any associated limitations reported in the literature. This may also provide more context for the section on Page 10, Row 193, that states issues around relapse in studies testing IGD interventions.

5. Page 9, Rows 179-182: The discussion here appears to shift to smartphone addiction, which may overlap with but does not always encompass addictive gaming behaviors. Please consider making a clear link between smartphone addiction and IGD in this section.

6. Page 16, Row 314: Clarifying how, where, and when participants were encouraged to practice independently could strengthen this section.

7. Page 16, Row 333-337, and Page 17, Row 338-347: In the previous section 2.6, it states that the control group will be led by a psychology instructor. Could you please clarify the specific role of the psychology instructor, given the audio-guided nature of the progressive muscle relaxation training? If the instructor will not be providing the progressive muscle relaxation training, please clarify this in section 2.6 and perhaps state what exactly they will be responsible for (e.g., participant management).

8. Page 21, Rows 385-386: Please clarify what the data collection is referring to here, along with the total number of reminder messages that will be sent per timepoint if a participant does not complete evaluations/follow-up.

9. Page 23, Row 397: This section mentions a pilot study of the current project. You could consider including some results and insights from the pilot study in the introduction section of the manuscript.

10. Page 27, Row 497: The manuscript raises an important point on dropout throughout the manuscript. If any strategies to reduce dropout have been planned, please discuss them in the manuscript. The following systematic review stated that the average follow-up retention rate was only 24% for internet-delivered interventions for internet-enabled addictive behaviors (inclusive of IGD) and discussed ways in which some studies addressed the risk of attrition: Park, J. J., King, D. L., Wilkinson-Meyers, L., & Rodda, S. N. (2022). Content and effectiveness of web-based treatments for online behavioral addictions: systematic review. JMIR mental health, 9(9), e36662.

11. Table 2: You could consider providing a brief description of the following for readers unfamiliar with these practices or stories: (i) story of the old well and pebbles, (ii) raisin exercise, (iii) “life is now” reading, (iv) mountain meditation practice, (v) bean counting exercise.

12. Table 3: I am interested in knowing how “average daily gaming time” (under “primary outcome”) and “average weekly gaming time” (under “others”) differ in terms of how they will be measured and interpreted. The table states that participants will self-record their average daily gaming time – will this also be the case for average weekly gaming time?

Minor issues:

1. Page 4, Row 57, and Page 6, Row 106: I recommend using “especially” instead of the shortened version.

2. Page 4, Rows 58-59: Please indicate if this statement is specific to IGD or other disorders.

3. Page 4, Row 60: Stating the three subcomponents in the abstract may improve clarity.

4. Page 4, Row 63: Since the acronym was introduced above, “Internet Gaming Disorder” could be omitted here.

5. Page 5, Row 80: Please add a space after the period. Similar issues have been noted in other areas (e.g., the period at the start of the paragraph on Page 12, Row 257, the repeated period on Page 14, Row 265, and a missing period in Table 2).

6. Page 6, Rows 95: Perhaps you could remove “which” here for better flow.

7. Page 6, Row 115: I believe this should be written as “functional magnetic resonance imaging (fMRI).”

8. Page 7, Row 120: Please clarify which part is controversial.

13. Page 9, Row 177: I would recommend changing “internet gaming addiction” to “internet gaming disorder” or “IGD” for consistency – please apply these changes to other sections of the manuscript that have used “internet gaming addiction.” Also, the reference (38) appears to not align with the statement it is supporting. I encourage you to double-check all in-text citations and the reference list.

14. Page 11, Rows 216-218: This sentence may be unnecessary in this section. If retained, it would be helpful if the sentence clearly reflects the primary and secondary aims stated in the abstract.

15. Page 11, Row 220: As this is the “study design and participants” section, please briefly state the design of the trial. Please feel free to transfer a modified version of the sentence from Page 14, Row 284-285 here.

16. Page 11, Rows 223-225: This sentence appears to be similar to the previous sentence. The two could be potentially combined to improve readability.

17. Page 11, Row 230: To maintain a formal tone, I would recommend removing “by the way.”

18. Page 12, Row 239: Please put a space before the in-text citation.

19. Page 15, Row 297: Please specify which behavior the brief mindfulness interventions have had moderate effects on executive function.

20. Page 15, Row 300: For consistency, I would recommend capitalizing the T in Figure 1.

21. Page 15, Row 304: It may be useful to specify if “school” refers to the university the study is recruiting from or multiple schools in a region.

22. Page 16, Row 313: Please specify if the post-session check-ins occur once a day or multiple times a day.

23. Page 17, Rows 353-354: I would recommend providing some examples of adverse events that may occur or directing the readers to section 2.12.

24. Page 17, Rows 360-361: Please consider stating some examples of monitoring mechanisms.

25. Page 24, Row 421, and Page 29, Row 538: I would recommend unbolding “This task will evaluate the updating component of executive” (Page 24, Row 421) and the letter “i” (Page 29, Row 538).

26. Page 26, Row 474: Please clarify if those involved in data analysis will be blind to the intervention allocation.

9. Page 37, Row 812: Please clarify if this is a thesis or peer-reviewed journal article.

27. Table 1: In the main body of the manuscript, please state the reason for the chosen cut-off point for the Internet Addiction Test and cite supporting studies. You could also perhaps remove the fifth inclusion criterion, as it is the opposite of the sixth exclusion criterion. Lastly, please provide the full names of the acronyms provided in the table (e.g., GAD-7).

10. Table 2: In the “post-class exercise (Ruixin app)” row, could you please clarify whether participants were instructed to listen to relaxation music? If applicable, please also indicate who provided this guidance and how it was communicated. Also, please remove the space after “meditation” in the Day 6 row.

11. Table 3: Please consider changing “others” to “Others” for consistency.

12. Figure 1: It may be helpful to provide an explanation of why primary and secondary outcomes were measured at both enrolment (-T2) and allocation (-T1).

13. Figure 2: The caption for this figure could remove the acronyms for BMT, PMR, and CON, as they are not included in the figure.

14. Figure 3: Please provide the unabbreviated term for HRV.

7. PLOS authors have the option to publish the peer review history of their article (what does this mean?). If published, this will include your full peer review and any attached files.

Reviewer #1: No

Reviewer #2: **Yes: **Kunru Song

Reviewer #3: **Yes: **Jennifer J. Park

---

## [Author Response · Author response to Decision Letter 0]

30 Dec 2024

Dear Editor and Reviewers,

We extend our sincere gratitude for the thoughtful feedback provided on our manuscript. We deeply appreciate the time and effort invested in reviewing our work. The insightful comments have been instrumental in significantly strengthening our research, leading to substantial improvements in both the methodology and presentation of our findings. The revised manuscript, thoughtfully incorporating all suggestions, is submitted for your kind consideration.

Reply

We have updated the manuscript and file names to comply with PLOS One's style requirements. Please see the resubmitted manuscript.

Thank you for stating the following financial disclosure: [This study was supported by the National Natural Science Foundation of China (NSFC) (Nos. 32060196, 82360271 and 82201597), Yunnan Fundamental Research Projects (202401AT070332), Joint Funds for Medical Specialization of KUST (KUST-KH2023002Y), Yunnan Ten Thousand Talents Plan Young and Elite Talents Project (YNWR-QNBJ-2018-027, YNWR-QNBJ-2018-056) and The innovation team of stress and disorder in nervous system in Yunnan Province (202305AS350011).]. Please state what role the funders took in the study. If the funders had no role, please state: "The funders had no role in study design, data collection and analysis, decision to publish, or preparation of the manuscript." If this statement is not correct you must amend it as needed. Please include this amended Role 

Reply

The financial disclosure was updated to include all the NSFC grant numbers (Nos. 32060196, 32460211, 82360271 and 82201597). The funders had no role in study design, data collection and analysis, decision to publish, or preparation of the manuscript. The financial support was provided in forms of research materials. The authors have declared that no competing interests exist. The statement is included in the cover letter.

Review Comments to the Author

Reviewer #1: Line 226: the randomization is unclear. State clearly how these 82 individuals with internet gaming addiction and 40 recreational gamers are allocated into the intervention and control group.

Reply

Thank you for your valuable suggestions. Allocation will be obtained via email from staff uninvolved in other aspects of the trial. Participants were stratified by gender and IGD severity before random assignment to either the experimental or control group using a random number generator. This ensured a balanced distribution across groups. See Section 2.5, “Group allocation”, lines 358-362.

Line 281: Several primary outcomes will be studied. Is the sample size calculated based on executive function sufficient to adequately support the analysis of the other primary outcomes?

Reply

Thank you for your valuable feedback regarding the sample size calculation. Our study encompasses several primary outcomes, including scale scores, behavioral measures, EEG, and HRV. The sample size calculation prioritizes the effect size of executive function, particularly from Stroop task designs, as this represents the most challenging outcome in terms of both effect size and variance. This approach ensures sufficient statistical power to analyze all primary outcomes. A detailed explanation of this rationale is provided in lines 325-334.

Line 292 for the progressive muscle relaxation training: To mention it will be held 7 days similarly with the intervention group.

Reply

Thank you for your suggestion regarding the description of the progressive muscle relaxation training. We appreciate your attention to detail. To clarify, we will explicitly state in the manuscript that the progressive muscle relaxation training will be conducted over a period of 7 days, similar to the intervention group. We have modified lines 365-366 to include this information: “Simultaneously, participants in the control group will receive progressive muscle relaxation (PMR) training, which will be conducted over a period of 7 days, consistent with the intervention group.” Thank you again for your helpful feedback.

Line 294: The sentence ‘following the intervention’ is to be omitted.

Reply

Thank you for pointing this out; it has been corrected.

Line 302: A separate subtitle to indicate intervention and control group respectively.

Reply

We have revised the relevant section to include distinct subtitles for each group, ensuring that readers can easily differentiate between the intervention and control conditions (lines 380, 415).

Information on the person who provide the training and do the data collection is to be provided.

Reply

Thank you for your valuable feedback. We have clarified the trainers for the intervention and control groups, as well as the individuals responsible for data collection in each group (lines 381-385, 416-419).

Page 22 Table 3: Information on whether all these inventories/questionnaires are in Chinese version and has been validated is to be provided.

Reply

Thank you for your insightful feedback. We have added references to support the information regarding the Chinese version of the questionnaires, including their sources and validation status (line 487, and Table3).

Data management on missing data if any is to be mentioned.

Reply

Thank you for your valuable suggestion. We have revised the data management section regarding any missing data in lines 564-571. We will incorporate the assumption of missing at random (MAR) and use multiple imputation techniques to enhance accuracy. Additionally, we will conduct sensitivity analyses to evaluate the impact of different missing data scenarios.

Line 476: Usually repeated measures ANOVA post hoc tests able to handle both within-group and between-group comparisons. Any reason why independent t test and paired t test are used?

Reply

Thank you for your helpful comments. We have clarified that repeated measures ANOVA is best suited for analyzing data collected across multiple time points. The sentence at line 605 has been revised for clarity: "Repeated measures ANOVA will be applied to analyze data collected at different time points across multiple sessions."

Line 478: The name of the effect size is to be provided.

Reply

Thank you for your valuable suggestion. Cohen’s d and partial eta-squared values will be used to measure primary and secondary outcomes in order to assess treatment impact (lines 607-608).

Line 478: Complete citation for SPSS 20.0 and R software including publisher name and version is to be provided.

Reply

Revised, see lines 605-606.

Line 478-479: Effect size with 95%CI to be stated.

Reply

Revised, see line 610.

The accepted statistically significant level is to be provided.

Reply

Thank you for your feedback. We have added the statement: “Data yielding a p < 0.05 will be considered statistically significant”. Please see line 609.

Line 496-499: To state clearly whether both intent-to-treat analysis and per protocol analysis will be conducted. If there are differs ‘n’ at various time points (due to attrition rates or missing data), using repeated measure ANOVA becomes problematic.

When performing repeated measures ANOVA with multiple time points, there is a risk of inflating the type I error rate (false positives) due to multiple comparisons. Thus, it is important to impose Bonferroni correction during post hoc testing to adjust the significance level for each comparison.

Figure 2: the analysis section requires revision.

Reply

Thank you for the constructive suggestions. We will conduct both Intention-To-Treat (ITT) and Per Protocol (PP) analyses to assess the intervention effects. The ITT analysis preserves random assignment to reduce bias, while the PP analysis focuses on participants who completed the intervention. We acknowledge that variations in sample size at different time points may complicate repeated measures ANOVA. We will implement this analysis cautiously, accounting for sample size changes to maintain statistical power. Moreover, to prevent inflating the Type I error rate during multiple comparisons, we will apply Bonferroni correction in our post hoc testing to adjust significance levels, ensuring the reliability of our findings, (lines 632-645). The analysis section of Figure 2 has been revised based on the text content.

References did not conform to the journal format.

Reply

We are grateful for your valuable input. Each reference has been carefully reviewed and updated as necessary.

Reviewer #2: The manuscript presents a well-conceived study protocol that is likely to contribute valuable insights into the impact of mindfulness on executive function in individuals with IGD. By exploring the effects of mindfulness training on executive function in individuals with IGD, the authors aim to bridge a gap in current intervention methods and provide new therapeutic insights. The manuscript clearly defines both primary (e.g., EEG data, executive function) and secondary (e.g., anxiety, stress) outcomes, ensuring that both cognitive and psychological dimensions are covered. This comprehensive evaluation can provide a holistic understanding of the intervention's impact on IGD.

I only have some minor concerns for this well-documented manuscript.

1. Longitudinal Follow-up Justification: While the study’s longitudinal design is commendable, the rationale for the choice of follow-up intervals (e.g., one week, one month, etc.) could be strengthened. Are these intervals based on previous findings regarding the duration of mindfulness effects? Providing a clearer justification for these time points could clarify the expected temporal effects of mindfulness training.

Reply

Thank you for your detailed and constructive feedback. We have clarified the rationale for our follow-up frequency selection:”This schedule is based on previous research showing that monthly follow-ups effectively monitor sustained benefits while balancing participant engagement and feasibility”. Supporting literature has been added. See lines 300-302.

2.Control Condition (Progressive Muscle Relaxation): The choice of progressive muscle relaxation (PMR) as a control condition is appropriate for differentiating between general relaxation and mindfulness-specific effects. However, the manuscript could benefit from a brief discussion on whether PMR is expected to influence executive function or gaming cravings to any degree, as this could affect the interpretation of results.

Reply

We are grateful for your suggestions. Accordingly, we have reviewed the relevant literature and provided additional clarifications.”Previous research observed slight gaming craving reduction with PMR, potentially due to a placebo effect. However, PMR group did not show significant changes in executive control network connectivity, and the difference from the mindfulness intervention group remained substantial. This highlights mindfulness' unique efficacy, as PMR uses a distinct relaxation mechanism”. See lines 422-427.

3. Terminology and Consistency: Ensure consistency in the terminology, such as using “Internet Gaming Disorder” (IGD) consistently instead of alternating with "gaming addiction" to prevent ambiguity.

Reply

Thank you for your detailed comments. We have carefully reviewed each inconsistent term and made the necessary revisions.

4. Future Implications: The manuscript could be further strengthened by a more explicit discussion of how the findings may be translated into clinical practice, particularly for interventions beyond university-based or app-based contexts.

Reply

Thank you for your valuable and insightful feedback. Based on your suggestions, we have supplemented the 'Strengths and Future Implications' subsection in the Discussion. Specifically, we have added content regarding ”The potential for brief mindfulness interventions could be integrated into various clinical settings, such as primary care clinics, community health centers, and mental health practices. For example, healthcare providers can incorporate brief mindfulness exercises into regular appointments, and community organizations can offer group sessions. Additionally, training programs for clinicians and educators can be developed to ensure they are equipped to deliver these interventions effectively. This approach would make the benefits of mindfulness more accessible to a broader population, including those with IGD, and support their executive function improvements in real-world settings”, see lines 684-692.

Reviewer #3: This manuscript presented an interesting RCT protocol that will examine the effectiveness of a brief mindfulness intervention on executive function in individuals that game (i.e., individuals with IGD and individuals without). I have provided comments on specific areas for improvement below.

Major issues:

1. Page 4, Rows 69-70: In the abstract, it may be beneficial to specify what you will be looking at in the EEG and what the behavioral metrics will specifically be measuring. Please also consider applying these changes to the main text of the manuscript (e.g., Page 21, Rows 370-371).

Reply

Thank you for your valuable suggestions. We have explicitly included the EEG and behavioral measures to be assessed in the abstract (lines 72-75) and have also made corresponding additions in the main text (lines 474-480).

2. Page 6, Row 113: I would recommend being slightly cautious of the phrasing here, as there is no consensus in the literature on the primary diagnostic characteristic of IGD. However, I agree that “excessive gaming behavior despite awareness of negative consequences” is one of the key characteristics of IGD.

Reply

Thank you for your constructive suggestions. We have revised the description to: “The primary behavioral characteristic of IGD is excessive gaming behavior despite awareness of negative consequences...” (line 123).

3. Page 7, Row 119: Given the focus on shifting and updating abilities throughout the manuscript, a brief description of each might be helpful to readers.

Reply

Thank you for your valuable feedback. We have added a brief description of shifting and updating; see lines 117-122.

4. Page 7, Rows 123: Before introducing the mindfulness sections, please provide more context on current IGD treatment by briefly discussing the most common intervention types for IGD, those that have shown promise, and any associated limitations reported in the literature. This may also provide more context for the section on Page 10, Row 193, that states issues around relapse in studies testing IGD interventions.

Reply

Thank you for your valuable suggestion. We appreciate your guidance on providing more context regarding current IGD treatment approaches. We have included a brief overview of the most common intervention types and summarized their strengths and limitations. Please see lines 230-235 for details.

5. Page 9, Rows 179-182: The discussion here appears to shift to smartphone addiction, which may overlap with but does not always encompass addictive gaming behaviors. Please consider making a clear link between smartphone addiction and IGD in this section.

Reply

Thank you for your helpful suggestions. We have added necessary information on the relationship between PSU and IGD, which is summarized as followsz:“Earlier studies suggested that the severity of problematic smartphone use (PSU) exhibits similar traits to other forms of internet overuse, such as IGD. Additionally, the high accessibility and diverse range of smartphone applications may contribute to this overuse”,Please see lines 212-215 for details.

6. Page 16, Row 314: Clarifying how, where, and when participants were encouraged to practice independently could strengthen this section.

Reply

Thank you for pointing this out. Regarding the after-class exercises, we have added the following clar

---

## [Decision Letter · Decision Letter 1]

18 Feb 2025

Protocol for a Randomized Controlled Trial to Enhance Executive Function via Brief Mindfulness Training in Individuals with Internet Gaming Disorder

PONE-D-24-33010R1

Dear Dr. Chen,

We’re pleased to inform you that your manuscript has been judged scientifically suitable for publication and will be formally accepted for publication once it meets all outstanding technical requirements.

Kind regards,

Metin Çınaroğlu

Academic Editor

PLOS ONE

Additional Editor Comments (optional):

Reviewers' comments:

Reviewer's Responses to Questions

**Comments to the Author**

1. Does the manuscript provide a valid rationale for the proposed study, with clearly identified and justified research questions?

Reviewer #1: Yes

Reviewer #2: Yes

Reviewer #3: Yes

2. Is the protocol technically sound and planned in a manner that will lead to a meaningful outcome and allow testing the stated hypotheses?

Reviewer #1: Partly

Reviewer #2: Yes

Reviewer #3: Yes

3. Is the methodology feasible and described in sufficient detail to allow the work to be replicable?

Reviewer #1: Yes

Reviewer #2: Yes

Reviewer #3: Yes

4. Have the authors described where all data underlying the findings will be made available when the study is complete?

Reviewer #1: Yes

Reviewer #2: Yes

Reviewer #3: Yes

5. Is the manuscript presented in an intelligible fashion and written in standard English?

Reviewer #1: Yes

Reviewer #2: Yes

Reviewer #3: Yes

6. Review Comments to the Author

You may also provide optional suggestions and comments to authors that they might find helpful in planning their study.

Reviewer #1: The authors have addressed the comments. No further comments. The manuscript is suitable for publication.

Reviewer #2: The authors made significant improvements on their manuscript. All my concerns have been adequately addressed.

Reviewer #3: Thank you very much for addressing my comments and providing thoughtful revisions. I have no further concerns and am pleased to recommend the manuscript for acceptance.

7. PLOS authors have the option to publish the peer review history of their article (what does this mean?). If published, this will include your full peer review and any attached files.

Reviewer #1: No

Reviewer #2: **Yes: **Kunru Song

Reviewer #3: **Yes: **Jennifer J. Park

---

## [Editor Report · Acceptance letter]

PONE-D-24-33010R1

PLOS ONE

Dear Dr. Chen,

I'm pleased to inform you that your manuscript has been deemed suitable for publication in PLOS ONE. Congratulations! Your manuscript is now being handed over to our production team.

Kind regards,

on behalf of

Dr. Metin Çınaroğlu

Academic Editor

PLOS ONE